# Clarifying terminology and definitions in education services for mental health users: A disambiguation study

Irina Pokhilenko[1], Mencia R. Gutierrez-Colosia[2]*, Luca M. M. Janssen[3], Silvia M. A. A. Evers[3,4], Agnes T. G. Paulus[3], Ruben M. W. A. Drost[3], Pilar Campoy-Muñoz[5], Judit Simon[6,7], Luis Salvador-Carulla[8,9]

1 Institute of Applied Health Research, Center for Economics of Obesity, Health Economics Unit, University of Birmingham, Birmingham, The United Kingdom, 2 Department of Psychology, Universidad Loyola Andalucía, Sevilla, Spain, 3 Faculty of Health, Department of Health Services Research, Care and Public Health Research Institute (CAPHRI), Medicine and Life Sciences (FHML), Maastricht University, Maastricht, The Netherlands, 4 Trimbos Institute Centre of Economic Evaluation & Machine Learning, Utrecht, The Netherlands, 5 Department of Economics, Universidad Loyola Andalucía, Sevilla, Spain, 6 Department of Health Economics, Center for Public Health, Medical University of Vienna, Vienna, Austria, 7 Department of Psychiatry, University of Oxford, Oxford, United Kingdom, 8 Health Research Institute, Faculty of Health, University of Canberra, Canberra, Australia, 9 Health Care Information Systems (CTS553), University of Cadiz, Cadiz, Spain

* menciaruiz@uloyola.es

## Abstract

In the wake of the mental health crisis in children and adolescents, the coordination of education and mental health services has become a global priority. However, differing terminologies and classifications across sectors, hinder effective comparison. The classification in education focuses mainly on outputs like qualifications or throughputs like teaching programs. This proof-of-concept study tested the applicability of a standard classification of health services, the Description and Evaluation of Services and DirectoriEs (DESDE), to evaluate education services for mental health users in the context of Spain and The Netherlands. It was conducted alongside the PECUNIA project, that sought to develop methods for the assessment of mental health costs and outcomes in different sectors. The study followed an ontoterminology approach involving: 1) identification of services from a predefined list of 46 resource-use items, 2) disambiguation of identified services with the DESDE, and classifying them as accurate, ambiguous, vague or confuse; and 3) external validation by an expert panel. The analysis was conducted at the level of type of resource, target population and care provision. From the initial list, only ten of the resources could be categorized as services using DESDE, and not activities, interventions or professionals. Only four of them (8,65%) were accurate across all disambiguation categories. Experts were unaware of terminology problems in classification of service provision in the education sector. Classifications and glossaries can clarify service naming, description and costing allowing comparative effectiveness analysis and facilitating cross-sectoral planning. This should be grounded in common methodologies, tools, and units of analysis.

**Data Availability Statement:** All relevant data are within the manuscript and its Supporting Information files.

**Funding:** The PECUNIA project has received funding from the European Union's Horizon 2020 research and innovation programme under grant agreement No 779292. This study also received financial support by a grant from the Carlos III Health Institute (PI18/01521). The funding agreement ensured the authors' independence in designing the study, interpreting the data, and writing and publishing the report.

**Competing interests:** The authors have declared that no competing interests exist.

# Introduction

In the wake of the mental health crisis in children and adolescents, the coordination of education and mental health services has become a global priority. Integrated youth services which rely on evidence-based models, are designed to provide effective care through a holistic approach [1]. This integration is part of a broader discipline known as service science that has emerged in response to the growing complexity of the service sector, particularly in more developed economies [2]. It is a multidisciplinary approach studying services from a holistic or whole systems perspective that incorporates ontology, i.e. classification and explanation of entities [3]. Person-centered service provision tends to be multi-sectoral in nature and include a multitude of service sectors such as, for example, education, healthcare, and social care [2].

In line with the systems thinking theory, understanding, complex systems require analysing their interrelationships and behaviours. Interventions influence the entire system, and likewise, an entire system perspective is needed to assess a complex intervention [4,5]. Moreover, every system, while unique in its structure, comprising elements like financing and governance, is not isolated. In fact, all systems are interrelated components of a larger social system. For example, improvements in health services may enhance educational outcomes by increasing student mental health, wellbeing etc. and conversely educational advancements can lead to better health awareness [6,7].

Therefore, cross-sectoral classification of services that considers the context of provision and the interconnected nature of systems, is essential. It may enable policymakers, to mitigate negative effects, maximize synergies, and assess the wider effects of interventions on the whole system [4]. Furthermore, there is empirical evidence from the health sector that the application of the whole systems approach in practice, leads to efficiency and improved outcomes through better coordination between the involved stakeholders [8–10].

While the nature of service delivery in specific sectors can differ, there are common challenges that pertain to most if not all sectors. One example of such a challenge is terminological variability. It constitutes a major source of systematic bias in service research, with the magnitude of this problem remaining largely unnoticed until very recently [11]. This has led to the increased interest in, and the relevance of, intersectoral international classifications that allow for meaningful comparison of services within and across different sectors, thereby supporting the application of the whole systems approach to the evaluation of complex interventions.

While several international classifications are widely used in the education sector, their primary focus has been on the comparability of outputs (i.e. outcomes such as educational qualifications) [12] or throughputs like educational programs and procedures [13]. Surprisingly, little information is available regarding the classification of inputs such as types of education services, settings and education teams using a standard classification of human service provision. The OECD classifies education services into core and ancillary types [14]. Core services include all direct instructional expenses like teacher's salaries, building maintenance, and administration. Ancillary services, though peripheral, focus on student welfare including meals, health services, transportation, and housing. Nevertheless, this classification lacks the tools needed for comparative analysis of services based on their features, within and across jurisdictions.

The International Standard Classification of Education (ISCED) represents another attempt to structure and organize education programs and educational qualifications by levels and fields [15]. However, it relies on country-level data implying important variabilities in classification and reporting. Also, there are programmes that span two or more ISCED levels making comparisons difficult.

There is a gap in the classification of provision of education services as current classifications do not establish common units of analysis that could be compared like-with-like. From an ecological perspective, several levels of units of analysis can be identified: macro-organizations (e.g. institutions, large foundations), meso-organizations (e.g. large schools), micro-organisations or services (e.g. small school, centres, courses), and nano-organisations that involve a dyad of users or groups of users and a provider delivering defined interventions and activities (e.g. a discipline within a school). Terminological variability and the lack of methods for classifying services according to common units of analysis lead to issues with service monitoring and planning. As mentioned above, this issue is relevant for many service sectors including the education sector and has been highlighted in previous studies focusing on school-based mental health interventions among other [16,17]. This issue was also raised in the impact analysis of the EdLINQ program for improving liaison between school services and child and adolescent mental health care in Queensland, Australia [18].

The Description and Evaluation of Services and DirectoriEs (DESDE) is an example of an international, intersectoral classification system valid for the evaluation of complex systems [19]. It originates from the field of mental health services research. In the past 25 years, it has been extended to incorporate other care sectors such as social [20], child and adolescent care, drug and alcohol, ageing, disabilities, long-term care [19,21]. DESDE has been validated and extensively used in the field of health services research and health economics for the comparison of resource utilization across different settings, efficiency analysis and the calculation of unit costs, among others [22]. Furthermore, DESDE is the only existing tool that provides local, bottom-up information that can be used across different sectors [5].

Building on the knowledge of classifications of health services [19] and its application to improve international comparability [23], this study aimed to implement a knowledge transfer from health to education services research. This study was designed as a proof of concept. That is the confirmation of the principle of the application of the emerging scientific knowledge, as the basis for subsequent activities, such as prototype development. Thus, we aimed to apply and test the applicability of a standardized classification system, i.e. DESDE, to the evaluation of education services relevant for mental health users in the context of two European countries, Spain and The Netherlands. The result of this study is expected to contribute to the understanding of the organization of education systems, identification of potential gaps in service provision, and facilitation of the application of the whole systems approach to the analysis of cross-sectoral service provision and complex interventions in the European context.

## Methods

The study was designed as a proof-of-concept to test the applicability of the DESDE classification system to the education sector. It aimed to assess the clarity of service terms in this sector using an ontoterminology approach. Meaning the study of disambiguation of technical and scientific terms using standard classifications [24]. This method helps identify precise definitions and relationships between terms, enhancing the retrieval of domain-specific information. It also aids in creating controlled vocabularies, classifications, and glossaries [24].

The study was conducted alongside the ProgrammE in Costing, resource use measurement and outcome valuation for Use in multi-sectoral National and International health economic evaluAtions (PECUNIA) project (https://www.pecunia-project.eu/). PECUNIA was a European-funded consortium aimed at developing new standardised, harmonised and validated methods and tools for the assessment of costs and outcomes in European healthcare systems from a whole-system perspective including health and social care, employment, education and justice sectors. Although methods and tools developed within the PECUNIA project were

meant to be generic, three mental health disorders (depression, post-traumatic stress disorder, and schizophrenia) were selected as illustrative examples. Building on the foundational goals of PECUNIA, this study's focus on disambiguation and classification was related to the critical objectives of definition and harmonization within the project [11].The initial list of education resource-use items was compiled from literature reviews conducted within PECUNIA. Once classified with DESDE, these services were used in PECUNIA for cost calculation.

## Education context in Spain and in the Netherlands

**Structure.** In Spain, education is mandatory from 6 to 15 years [25]. Pre-school education while not compulsory, is available for children from birth to 6 years, and is divided into nursery school (0–3 years) and infant school (3 to 6 years) [25]. OECD reports indicate that in Spain 95% of 3 years old and 97% of 4 years old are enrolled in pre-school [26]. Education continues with primary school from ages 6 to 12, followed by secondary education until age 15. Post-secondary options such as upper secondary education or vocational training are available for 16 to 18 years, but are not compulsory [25]. Special education is organized in three sections: special early childhood education, adapted compulsory basic education, and training programs for transition to adult life from 16 to 20 years old, and it is not compulsory [27].

In the Netherlands, school attendance is compulsory from 5 to 16 years old, and for those between 16 and 18 who have not yet met basic qualifications [14]. Pre-school education is not compulsory [28]. Children typically start primary school at the age of 4 and transition to secondary education at 11–12. Secondary school education is structured into three distinct tracks [28]: pre-vocational education lasts four years and prepares students for entering vocational schools [29]; higher general continued education takes five years and prepares students to enter a university of applied sciences; preparatory scientific education lasts six years and prepares students to enter an academic university [30]. Special education is delivered in separate special schools and is organized into four clusters to address specific needs: visual impairments, hearing or speech impairments, physical or mental handicaps, and mental disorders and behavioural problems [31].

**Governance.** In Spain, the Ministry of Education, Culture and Sport governs the education sector. Individual schools and budget are managed by the autonomous communities with guidelines set by the central government. Public schools receive government subsidies and must adhere closely to Ministry requirements. Private schools which may be subsidised or not, also depend on and are authorised by the regional government, but they have their own management. Although Spanish universities operate independently from the government, they are regulated and monitored by the national and regional quality and accreditation agencies [26].

In the Netherlands, the Ministry of Education, Culture and Science, governs the education sector, setting general rules while allowing institutes freedom in their teaching methods [32]. Primary and secondary schools are further governed by school boards, with several schools falling under one board. The quality of education is monitored nationally by the Inspectorate of Education, and funding for schools is allocated based on the number of students; with additional support for disadvantaged students or students with special needs [33]. Both public and private schools can receive government funding if they meet statutory requirements. Vocational and higher education institutes operate as separate legal entities and receive funding through a combination of tuition fees and government grants, which are based on the institution's performance. Tertiary educational institutions are governed by boards that formulate and implement strategies within the government's framework [33].

**Key professionals.** In Spain, the main reference professional in public schools is the teacher/educator. They must complete four years of university studies and pass an official

exam. Experience and training provide points that allow them to move between institutions. Special education adopts, the multitrack model described by Nash and Norwich [34], where all teachers receive generic training followed by specialized training therapeutic pedagogy or hearing and language [35]. In the private sector, a master's degree is required in secondary though the official exam is not.

In the Netherlands, primary school teachers need a four-year degree from either a university of applied sciences or a regular university. Secondary school teachers, are differentiated by the level of education they can teach, depending on whether they have a first degree (lower education levels e.g pre-vocational education) or second degree (advanced educational levels e.g. preparatory scientific education). Teaching assistants, that support the delivery of education at primary and secondary schools, must complete a vocational education degree [36]. In special education, although the level might be lower compared to regular education, the pedagogical requirements are more stringent. Higher education professionals are often required to have a degree from a university of applied sciences or a regular university, along with a university teaching qualification, though there are no specific statutory requirements [36].

## Tools: Description and Evaluation of Services and DirectoriEs (DESDE)

The DESDE system constitutes a unique approach for the standard description and classification of services across sectors [19,21]. It first identifies minimal organizational units with temporal stability arranged for delivering care, called Basic Stable Inputs of Care (BSIC). Second, the main activity of the BSIC is described through the 'Main Type of Care' (MTC) or code. There are 108 codes subdivided into six main branches of care (Residential, Day, Outpatient, Accessibility, Information, Self-help and Voluntary). DESDE provides a multiaxial system that contains information on the following aspects of a service or a BSIC: a) the sector cluster (health, social, education, employment, justice); b) the target population, Including the age (e.g. CX–children and adolescents), gender, and the diagnosis group for which the service is intended. Code is based on the International Classification of Diseases 10[th] edition (ICD-10) [37] (e.g. F7-F8—Mental retardation, disorders of psychological development), and the International Classification of Functioning, Disability and Health (ICF) [38]; c) a code of the main service function or type of care and its additional qualifiers. This code is the core component of DESDE (Fig 1). Two DESDE axes were used for the disambiguation study, target population and service type of care. In the example presented in Fig 1, the code thread refers to a special education needs class in a school. In this proof of concept study, the rationale for applying DESDE classification arose from the need to create a standardized framework for describing and assessing education services in a comparable manner. DESDE was originally developed for use in health and social care sectors. Its methodological rigor made it ideal to bring similar benefits of standardization to the education sector.

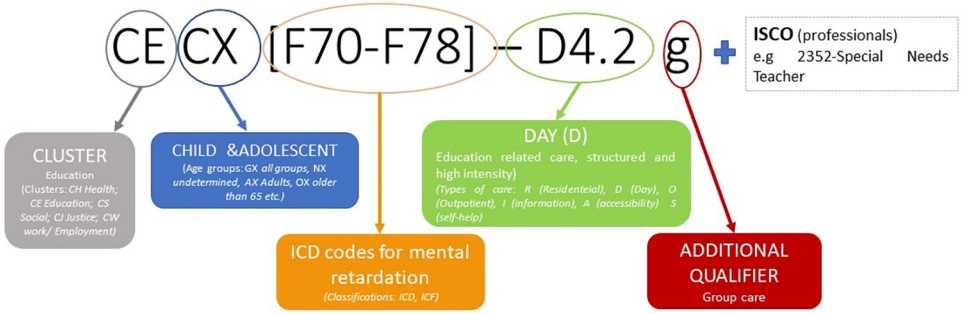

**Fig 1. DESDE multiaxial code structure.**

## Procedure

This ontoterminology study followed a multistep process to assess the clarity of terms in education services, and involved three working groups:

**Step 1—service identification.** Group A comprised members from Maastricht University (IP, LJ, SE, AP, RD). They undertook peer-reviewed and grey literature searches, as well as multi-national expert surveys that took place in six European countries prior to this study, as part of the PECUNIA project [39]. The aim was to build a list of resource-use items in the education sector along with their respective definitions relevant for mental health economic evaluations [39]. These identified services in the education sector, along with services in other sectors [40], formed the basis for developing new costing instruments such as the Resource Use Measurement (RUM) instrument, Reference Unit Cost (RUC) Templates, and RUC Compendium [41–43].

**Step 2 –disambiguation.** Group B included two experts in disambiguation using DESDE (MR, LS) that analysed the list compiled by group A. In terminology, disambiguation is the process of making something clear and resolving ambiguity and vagueness of terms [44]. Ambiguity refers to the terms that can be interpreted in more than one way because of imprecise definitions. Vagueness refers to the terms that are not sufficiently specified and therefore cannot be translated into a code. Our team established a threshold for differentiating ambiguity and vagueness in the analysis of health services using DESDE [44,45]. A service that could be assigned two codes of the reference classification system was considered ambiguous, whilst a service that could be assigned more than two codes, was categorized as vague. Confusion in this study refers to the terms whose definition did not allow classification. Items were categorized into 'accurate', 'ambiguous', 'vague', or 'confusing' based on descriptions. The process involved two levels:

- 1. Defining the unit of analysis (type of resource): group B reviewed the general list and ensured that the items and their descriptions corresponded to the same unit of analysis (commensurability), in this case "services" and not to other units of analysis such as interventions, professionals, or outcomes. The standard description of services provided in DESDE was used to ensure that items could be compared like-with-like. Items classified as non-services were excluded from further disambiguation.

2. DESDE classification: group B assessed information about the target population—including age and diagnosis group-, as well as the type of care provided in the services. This enabled the assignment of prototype DESDE codes and additional qualifiers.

**Step 3- external validation.** To finalize the disambiguation process and evaluate the suitability of DESDE for classifying education services, group C was consulted. This group consisted of a panel of external experts recruited from the current study's co-authors' professional network using purposive sampling (two from Spain, and three from the Netherlands). It included two academics, one public agent from the regional Ministry of Education and two third-sector professionals (AG, AR, TS, IO, MS). Their primarily role was to validate the services coded by group B, using their deep knowledge of the sector.

Consultation involved individual semi-structured interviews. Experts were guided through a series of generic inquiries about education within the contexts of Spain and The Netherlands. Interview questions were pre-defined by the co-authors of the current study (S1 Table). Experts critically reviewed the comprehensiveness and accuracy of the service list and verified the correctness of the assigned codes, taking into account the educational realities in both countries.

## Results

### Step 1 –identification

The item list for disambiguation contained 46 tentative services or resources, 38 items out of which were identified in the review of literature and expert surveys, and eight items were extracted from the PECUNIA RUM instrument [41]. For the full list of items, please refer to S2 Table. An additional listing of Spanish services and programmes was identified and provided in S3 Table. This list did not undergo disambiguation inasmuch as it was not included in the official listing of PECUNIA services, but it was reviewed by the Spanish experts.

### Step 2—disambiguation

**Level 1—unit of analysis (commensurability).**   In the first step of disambiguation, ten items (22%) were classified as accurate and included in the subsequent disambiguation steps. Thirty-five items (76%) were classified as confusing, because they did not refer to services but rather to procedures ('special needs diagnostics'), professionals ('liaison teacher') or consequences ('low school adaptation/competence') and were therefore excluded from the subsequent disambiguation steps. One item (2%), 'student counselling', was excluded because it was not clear what was understood by counselling in this term and admitted very different units of analysis. Please refer to S2 Table for more details.

**Level 2—DESDE classification.**   Ten items, initially classified as accurate in terms of unit of analysis and thus defined as "services" were further assessed for clarity concerning the target population. Five items were classified as accurate (50%). Two items were classified as ambiguous (20%) these included 'special education services' and 'student transport to special education facility' where both primary and secondary students could be admitted. Another two items were classified as vague (20%) because they lacked specific descriptions of age or diagnosis groups ('special education day school' and 'day school'). One item was classified as confusing (10%) due to an incoherent description of the target population; 'higher education school (university, college, vocational school)' could include not only adults, but also adolescents. Please refer to Table 1 for more details.

Regarding service type, eight items were classified as accurate (80%). One item, 'education support in a private setting' was classified as ambiguous (10%) due to it having two different codes, with the second code varying by the frequency of care provided from high (O8) to low (O10). Another item, 'special education services' was classified as vague (10%), because it required a wide range of different codes. For more details, please refer to Tables 1 and 2.

### Step 3—expert validation

Initially, the five experts from the panels (group C) were unaware of any terminology problem in the classification of educational services. Only one expert from Spain highlighted potential problems when comparing services between regions in the country. Experts considered 'special education services' and 'higher education school' umbrella terms that could refer to a variety of settings and discarded them as they were too broad to be useful for an operational classification. Regarding special education, they informed that in recent years there has been a shift towards a more inclusive school environment both in Spain and The Netherlands. Students with special education needs are educated in regular schools with additional support if required. For example, in the Netherlands regular schools receive additional funds to support students with special needs. Furthermore, partnerships between regular and special education schools facilitate the integration of well-performing students into regular schools. in Spain there were special centres for students with autism spectrum disorders and intellectual

**Table 1. Service description and disambiguation with DESDE classification (step 2).**

| Service name | Service description | Prototype DESDE code(s) | DESDE Description (target + code) | Disambiguation category |
|---|---|---|---|---|
| Boarding school | A small scale school offering regular education during the day as well as flexible boarding and full boarding facilities | CC [X]-R8.2 D4.2gq CA [X]-R8.2 D4.2gq | TP: Children (CC) adolescents (CA), non diagnosis DESDE code: Residential non-acute, 24h non medical support facilities for temporal stays longer than one month (R8.2) with a significantly higher length of stay than other R8.2 services (q); Non-acute day care facilities for education related care, available the equivalent of 4 half days/week (D4.2) for groups (g) | TP: accurate DESDE code: accurate |
| Day school | A day school providing regular education (language, math, history) to groups of children | CC [X]-D4.2g CA [X]-D4.2g | TP: Children (CC) adolescents (CA), no diagnosis •DESDE code: Non-acute day care facilities for education related care, available the equivalent of 4 half days/week (D4.2) for groups (g) | TP: vague (age groups not specified) DESDE code: accurate |
| Education support at your place of study (e.g. tutoring, additional lessons) | Additional educational support provided as an outpatient service at a regular primary/secondary school to students with learning difficulties (often referred to as remedial teaching) | CC [F81.9]-O8.2 or O9.2 or O10.2 CA [F81.9]-O8.2 or O9.2 or O10.2 | •TP: Children (CC) adolescents (CA), learning difficulties •DESDE code: Outpatient non-acute non-mobile facilities providing non-health related care more than three times/week (O8.2), once in a fortnight (O9.2) or once per month (O10.2) | TP: accurate DESDE code: accurate (all codes belong to the same type of care only frequency differences) |
| Education support in a private setting (e.g. private tutoring) | Additional educational support provided as an outpatient service in a private setting to students with learning difficulties (often referred to as remedial teaching) | CX [F7-F8]-O5.2.1 CX [F7-F8]-O8.2 or O9.2 or O10.2 | •TP: Children (CC) adolescents (CA), mental retardation (F7-F8) •DESDE code: Outpatient non-acute mobile facilities providing social care three to six day/week (O5.2.1); Outpatient non-acute non-mobile facilities providing non-health related care more than three times/week (O8.2), once in a fortnight (O9.2) or once per month (O10.2) | TP: accurate DEDE code: ambiguous |
| Higher education school (university, college, vocational school) | A high-level educational institution offering upper-secondary or tertiary education services to groups of students | AX [X]-D4.2g | •TP: Adults (AX), no diagnosis •DESDE code: Non-acute day care facilities for education related care, available the equivalent of 4 half days/week (D4.2), for groups (g) | TP: confusing (college in some countries is similar to high school, age group not clear) DESDE code: accurate |
| Night school | Adult learning school that holds classes in the evening or at night to accommodate people who work during the day | AX [X]-D4.2g | •TP: Adults (AX), no diagnosis •DESDE code: Non-acute day care facilities for education related care, available the equivalent of 4 half days/week (D4.2), for groups (g) | TP: accurate DESDE code: accurate |
| Special education boarding school | Facility providing care, support, education and employment to people of all ages with severe (intellectual) disabilities and complex behavioral and / or psychiatric problems. This facility offers special education during the day for groups of children and young people (max 16) who experience a structural limitation in their educational participation due to their behavioral disabilities or psychiatric problems | CC [F7-F8]-R8.2 D4.2gq CA[F7-F8]-R8.2 D4.2gq | •TP: Children (CC) adolescents (CA), mental retardation (F7-F8) •DESDE code: Residential non-acute, 24h non medical support facilities for temporal stays longer than one month (R8.2) with a significantly higher length of stay than other R8.2 services (q); Non-acute day care facilities for education related care, available the equivalent of 4 half days/week (D4.2) for groups (g) | TP: accurate DESDE code: accurate |
| Special education day school | A day school providing education and guidance for pupils with special educational needs | CC [F7-F8, ICF]-D4.2g CA [F7-F8, ICF]-D4.2g | •TP: Children (CC) adolescents (CA), mental retardation (F7-F8) •DESDE code: Non-acute day care facilities for education related care, available the equivalent of 4 half days/week (D4.2) for groups (g) | TP: vague (Can include other diagnosis besides mental retardation or developmental disorders) DESDE code: accurate |

*(Continued)*

**Table 1.** (Continued)

| Service name | Service description | Prototype DESDE code(s) | DESDE Description (target + code) | Disambiguation category |
|---|---|---|---|---|
| Special education services | Special education for children who are not able to be supported in regular school classes concerning personal developments and achievements | CC [F7-F8]-D4.2<br>CA [F7-F8]-D4.2<br>CC [F7-F8]-O8.2 or O9.2 or O10.2<br>CA [F7-F8]-O8.2 or O9.2 or O10.2<br>CC [F7-F8]-S1 or S2<br>CA [F7-F8]-S1 or S2 | •TP: Children (CC) adolescents (CA), mental retardation (F7-F8)<br>•DESDE code: Non-acute day care facilities for education related care, available the equivalent of 4 half days/week (D4.2); Outpatient non-acute non-mobile facilities providing non-health related care more than three times/week (O8.2), once in a fortnight (O9.2) or once per month (O10.2); Facilities that provide support, self-help or contact with unpaid staff, non-qualified (S1) or qualified (S2) | TP: ambiguous (access criteria is not defined for diagnosis or for age (primary, secondary) DESDE code: vague |
| Student transport to special education facility | Services aimed at transporting students to special education facility | CX [F7-F8]-A2 | •TP: Children and adolescents (CX), mental retardation (F7-F8)<br>•DESDE code: Accessibility to care. Mobility provided by other transport than ambulance (A2.1.2) | TP: ambiguous (Access criteria is not defined for age (primary, secondary) DESDE code: accurate |

TP: Target population.

disabilities among other conditions, but the trend in the last decades has been to offer universal education in regular schools with specific programmes for the students with special needs. Only individuals with more severe disabilities are typically accommodated in 'special education day schools'. Experts indicated that this distinction should be reflected in the DESDE code axis of the target population. Therefore, the ICF severity code (ICF3), was added in this case. 'Boarding school', 'special education boarding school', while no longer prevalent in public systems, were considered accurate in their definition and code. Similarly, 'day school' and 'Student transport to special education facility' were considered appropriate.

When asked about the comprehensiveness of the list of services, experts mentioned others not included in the more mental health-specific PECUNIA list. In Spain, teacher training centres offer voluntary and free training organized by teachers/educators who specialize in this area. This training represents an important part of the education budget. Temporary linguistic adaptation classrooms assist foreign students in learning Spanish, particularly in centres with high immigrant populations like temporary shelters. Sign language interpreters aid children with hearing difficulties. Schools can also request reinforcement, orientation, and support plans for students with specific learning difficulties, delivered after school. Furthermore, there are teachers providing domiciliary education for children with long term illnesses.

**Table 2. Disambiguation rates.**

| Level | Accurate | Ambiguous | Vague | Confusing |
|---|---|---|---|---|
| L1. Unit of analysis (n = 46) | n = 10<br>22% | n = 0<br>0% | n = 1<br>2% | n = 35 (not services but interventions, activities or professionals)<br>76% |
| L2. DESDE target population (n = 10) | n = 5<br>50% | n = 2<br>20% | n = 2<br>20% | n = 1<br>10% |
| L2. DESDE service type (n = 10) | n = 8<br>80% | n = 0<br>0% | n = 2<br>20% | n = 0<br>0% |
| Services accurate at all levels | N = 4<br>8,65% | | | |

In the Netherlands, there are day care facilities aimed at providing learning basic life skills to people who dropped out of school. These services are not allowed to provide education; but they are often located within special education schools. Experts indicated that the type of funding (private or public) should be added to the DESDE code.

Finally, there are specific resources both in Spain and the Netherlands for juvenile penitentiary centres and for the children and adolescents undergoing treatment in the hospital. Following the DESDE hierarchical system, these services are classified respectively as justice and health services that provide education. The list of services, prototype codes, and comments from the expert interviews are listed in Table 3 below.

**Table 3. Services, prototype codes and comments from the expert interview.**

| Service name | Prototype DESDE code(s) | Comments from the expert from Spain | Comments from the experts from the Netherlands | DESDE code(s) |
|---|---|---|---|---|
| Boarding school | CC [X]-R8.2 D4.2gq CA [X]-R8.2 D4.2gq | 'Disappearing in Spain' | 'Do not exist in the Netherlands' | Ok Not available |
| Day school | CC [X]-D4.2g CA [X]-D4.2g | | | Ok |
| Education support at your place of study (e.g. tutoring, additional lessons) | CC [X]-O8.2 or O9.2 or O10.2 CA [X]-O8.2 or O9.2 or O10.2 | 'Equivalent to special education support' | | CX [F7-F8] O8.2 Suggestion to include code to indicate special services delivered in regular schools |
| Education support in a private setting (e.g. private tutoring) | CX [F7-F8]-O5.2.1 CX [F7-F8]-O8.2 or O9.2 or O10.2 | | | CX [F7-F8] O8.2 Suggestion to include code to indicate special services delivered in regular schools |
| Higher education school (university, college, vocational school) | AX [X]-D4.2g | 'Universities are managed by a different organism, the Ministry of universities. Typically for people over 18 years old'. | 'There is no lower age limit for enrolment. Entry is based on secondary education degree'. | Umbrella term |
| Night school | AX [X]-D4.2g | 'For people over 18 years old or people with 16 years but with a job contract. It is possible to study basic education (graduate on secondary education) or high school/baccalaureate. Some high schools offer this education at evenings, and there are also specific centres. Teachers/educators are the same of the regular education' | 'This is called adult education in the Netherlands. These services are coordinated by the government. Some services can be private. Professionals working in adult education need the same education as those working in regular schools. Adult education is organized in specific centres that need to obtain a license'. | AX [X]-D4.2g Suggestion to include code to differentiate day and evening teaching |
| Special education boarding school | CC [F7-F8]-R8.2 D4.2gq CA[F7-F8]-R8.2 D4.2gq | 'Boarding schools are almost non-existent in Spain. At least there are not centres publicly funded. Professionals are educators. Currently there are school residences for children that live in remote or rural areas and need accommodation to attend school. There are teachers available during the day in these residences to help children to organize their studies, but they don't actually teach'. | 'There are special education schools with boarding services, but these are always privately funded. They provide services for extremely gifted children and use very specific teaching methods. Or there are privately funded (also from health budget) schools for students with mental health issues'. | Ok Not publicly available |
| Special education day school | CC [F7-F8, ICF]-D4.2g CA [F7-F8, ICF]-D4.2g | 'Equivalent to Special education centre. Only for very severe learning disabilities' | 'Only those with strong learning disabilities who cannot fit in regular education, go to special education schools'. 'Within regular schools, there are funds for students with special needs. Some regular schools work together with special education schools to slowly integrate more talented students into regular schools'. | CX [F7-F8, ICF3] D4.2g |

*(Continued)*

**Table 3.** (Continued)

| Service name | Prototype DESDE code(s) | Comments from the expert from Spain | Comments from the experts from the Netherlands | DESDE code(s) |
|---|---|---|---|---|
| Special education services | CC [F7-F8]-D4.2<br>CA [F7-F8]-D4.2<br>CC [F7-F8]-O8.2 or O9.2 or O10.2<br>CA [F7-F8]-O8.2 or O9.2 or O10.2<br>CC [F7-F8]-S1 or S2<br>CA [F7-F8]-S1 or S2 | 'Umbrella term for all the possible services'.<br>•Special education centres: CX [F7-F8, ICF3] D4.2g<br>•Special education classrooms in regular school: CX [F7-F8, ICF2] D4.2gs ('p').<br>•Special education support in the regular classroom: CX [F7-F8] O8.2s.<br>•Children are evaluated by local educational guidance teams (Equipos de Orientación Educativa de zona EOE). These teams make the assessment and identification of children with special education needs, and facilitate an educational response to children and primary schools. | 'Very broad, but correct description'. | Umbrella term, not possible to assign one code |
| Student transport to special education facility | CX [F7-F8]-A2 | 'Transport, extracurricular activities and meals function similarly, only families with very special socioeconomic needs receive it for free, the rest pay specific amounts based on their incomes'. | 'Generally, transport for students is not regulated. Municipalities are responsible for special needs services. Parents can ask for transport for special needs schools, pay for it themselves or get a subsidy'. | OK |

## Discussion

This study on educational services relevant to mental health, presents the first analysis of terminological unclarity and highlights the need for a standard international terminology and classification of services in this sector. The rationale for applying DESDE classification arose from the need to create a standardized framework for describing and assessing education services in a comparable manner from a whole-system and cross-sectoral perspective [44]. The magnitude of the terminological problem in the education sector is at least as relevant as it is in the health and social sectors. From the original list of 46 resource-use items, only ten were typified as actual services (resource inputs) while other terms referred to professionals, interventions (throughputs), and consequences (outputs). Five services were accurate at target population level and eight were accurate at the level of service type (DESDE code). Only four services were accurate at all levels of disambiguation (8,65%) namely 'boarding school', 'special education boarding school', 'education support at your place of study' and 'night school'. This ambiguity reflects the complexity and diversity of education services and the challenges inherent in defining and categorizing them. Accurate classification informs policy development by providing policymakers with a clear understanding of the types of services available and the population they serve. This allows for a more effective resource allocation and collaboration across sectors.

Most services were included in the 'day' typology of care, with the main code of 'education related, structured and high intensity care, delivered for groups of people' (D4.2g). However, DESDE has not been previously applied to assess the education sector, only to specific services providing education care for mental health users in the health and social sectors [46].

Collecting data using a top-down approach from sources such as administrative databases or professional boards, where data is interpreted through a national prism, may imply simplifications of the reality of the local system. Without clear terminology, applying national data to the local level could lead to misunderstandings and likely failures of any intended changes or interventions in the real world. Evidence about local conditions is important for assessing resource availability. A bottom-up approach that includes a comprehensive and systematic

description of the context and knowledge at the service delivery level is essential to complete national data for evidence-informed research and policy [5].

Expert consultation was in this sense crucial for nuanced service descriptions, appropriate code assignments, and gaining deeper insights into the national education systems from a bottom-up perspective. Their input, based on their contextual background, contributed significantly to refining the outputs of the study and enhance the reliability of the findings: 'boarding school' and 'special education boarding school', were considered correctly classified despite being residual and absent from the public sector of both countries. Additionally, 'day school' and 'student transport to special education facility' were accurate at the level of DESDE code, although the latter lacked specificity regarding target population due to unspecified age groups. 'Education support at your place of study', and 'education support in a private setting' received high intensity codes within the provided range. Finally, 'special education day school' required a code for severity. 'Higher education' and 'special education', were considered umbrella terms and not useful for classification.

Suggestions made by experts for adapting the DESDE system to the education sector included the future development of new codes to differentiate day and evening school, reflecting differences in operational contexts such as hours, staffing, or resources. As well as differentiating regular education from special services provided within regular school premises. However, the information obtained in the study is established as the first step for the development of a classification of education services that will enable comprehensive mapping of the education sector in specific geographical areas. This standardized framework will facilitate systematic data collection and better comparison on the availability of services in a region at every level of the ecological system (micro, meso and macro).

Integrating international classifications has demonstrated a potential for disambiguation and therefore improving the comparability of resources across settings [20,44]. The combined use of complementary classification systems of services, interventions and professionals has proved its value in health (e.g. psychotherapy) [44] and social care (e.g. case management) [45]. In education, the description of service provision using DESDE can be combined with information on programs and qualification levels from ISCED and the types of professionals involved (ISCO-International Standard Classification of Occupations [47]. Thus, a more comprehensive overview of the education sector could be achieved in evaluating a primary school for special education, DESDE could provide insights into the facilities structured day care (i.e. D4.2 day care for a fixed number of hours/days); ISCED could clarify the level of education offered such as primary education focusing on basic skills like reading, writing and mathematics (ISCED level 1); and ISCO could detail the roles of professionals, like special education teachers (ISCO code 2340). This approach would enhance comparability and support evidence-informed decision-making [13].

## Limitations

The findings of this study must be seen in light of some limitations. First, this study is a proof of concept for the development of an ontology-based classification of education services and its related terminology. It has been based on the knowledge transfer from health and social service research to the education sector. The DESDE classification system was developed for and has traditionally been used in health and social services research with corresponding wording and rationale. Nonetheless, the DESDE classification system has proven to be valid for evaluating care services in other target populations like old age and people with addiction and disabilities [22]. Second, this study relies on the list of specific education services related to mental health, as an inclusion criterion described in the PECUNIA project. Therefore, this study does

not provide the full terminology and nomenclature of services provided in the education sector, although the systematic review and the expert survey were international and comprehensive [39]. Third, a limited number of experts was consulted in each country, which could have provided a limited perspective on the applicability of DESDE to classifying education services in Spain and in the Netherlands.

## Implications for further research

Comparison of service provision, regardless of the sector should be grounded in standard methodologies. Using well-defined units of analysis ensures commensurability, i.e. like-with-like comparisons within or across areas [48]. Classifications and glossaries help overcome terminological bias preventing ambiguity and vagueness in the naming and description of existing services as well as avoid overshadowing duplication of services while recognizing the actual diversity of services [49]. This study presents a knowledge transfer from the health to the education sector by applying the DESDE system and lays out methods for applying this approach in countries other than Spain and the Netherlands.

As the first step, this proof of concept is important to set the ground for conducting a comprehensive mapping of an area and to study the feasibility and expanding the classification to encompass the entire education system. Standardized data enables policymakers and educators to make decisions based on robust evidence thereby enhancing effective service planning and delivery. Additionally, adopting a standardized classification for a whole system approach, would facilitate deeper integration with other sectors like healthcare, ensuring that policies and practices are interconnected across sectors, enhancing evidence-informed decision-making, and optimizing allocation of resources.

## Supporting information

**S1 Table. Questions of the expert interview.**
(DOCX)

**S2 Table. Commensurability assessment (unit of analysis).**
(DOCX)

**S3 Table. Spanish services and programmes.**
(DOCX)

## Acknowledgments

Authors want to acknowledge the contribution of Ana Gimenez Ciruela, PhD, Counselor at the Regional Ministry of Education of Andalucía (Spain); Ana Rodriguez Meirinhos, Department of Education, Universidad Loyola Andalucía (Spain); Prof. Trudie Schills, Professor of Economics of Education at Maastricht University (The Netherlands); Ingrid Ottenheijm, Director of the Special Education Partnership in South Limburg (The Netherlands); Mrs Milou Samuels, Teach and Talent Coach for gifted children at Bernardinus College (The Netherlands), and the PECUNIA group for their contribution to the development of the list of services.

## Author Contributions

**Conceptualization:** Irina Pokhilenko, Mencia R. Gutierrez-Colosia, Luca M. M. Janssen, Pilar Campoy-Muñoz, Judit Simon, Luis Salvador-Carulla.

**Data curation:** Irina Pokhilenko, Luca M. M. Janssen, Pilar Campoy-Muñoz.

**Formal analysis:** Luca M. M. Janssen, Judit Simon.

**Funding acquisition:** Judit Simon.

**Investigation:** Irina Pokhilenko, Mencia R. Gutierrez-Colosia, Luca M. M. Janssen.

**Methodology:** Irina Pokhilenko, Mencia R. Gutierrez-Colosia, Luca M. M. Janssen, Judit Simon, Luis Salvador-Carulla.

**Resources:** Mencia R. Gutierrez-Colosia.

**Supervision:** Silvia M. A. A. Evers, Agnes T. G. Paulus, Ruben M. W. A. Drost, Luis Salvador-Carulla.

**Validation:** Silvia M. A. A. Evers.

**Writing – original draft:** Irina Pokhilenko, Mencia R. Gutierrez-Colosia, Agnes T. G. Paulus, Ruben M. W. A. Drost, Pilar Campoy-Muñoz, Judit Simon, Luis Salvador-Carulla.

**Writing – review & editing:** Irina Pokhilenko, Mencia R. Gutierrez-Colosia, Silvia M. A. A. Evers, Judit Simon, Luis Salvador-Carulla.

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
