## [Decision Letter · Decision Letter 0]

1 Mar 2024

PONE-D-23-36992Clarifying Terminology and Definitions in Education services for Mental Health Users: A Disambiguation StudyPLOS ONE

Dear Dr. Gutierrez-Colosia,

Thank you for submitting your manuscript to PLOS ONE. After careful consideration, we feel that it has merit but does not fully meet PLOS ONE’s publication criteria as it currently stands. Therefore, we invite you to submit a revised version of the manuscript that addresses the points raised during the review process.

**ACADEMIC EDITOR: **

Editor’s comments:

After a thorough examination of your manuscript, it is apparent that your work exhibits significant potential. I kindly request some minor revisions to further enhance the manuscript. These suggestions are tailored to pinpoint specific areas where refinement could enhance overall clarity and amplify the impact of your work. The detailed suggestions are provided below, and I kindly ask you to respond to both the editor and the reviewers. Please address each suggestion individually in a separate file and highlight the modifications made in the manuscript. Your attention to these revisions is greatly appreciated.

Editor’s comments:

After a thorough examination of your manuscript, it is apparent that your work exhibits significant potential. I kindly request some minor revisions to further enhance the manuscript. These suggestions are tailored to pinpoint specific areas where refinement could enhance overall clarity and amplify the impact of your work. The detailed suggestions are provided below, and I kindly ask you to respond to both the editor and the reviewers. Please address each suggestion individually in a separate file and highlight the modifications made in the manuscript. Your attention to these revisions is greatly appreciated.

**Abstract**

The abstract, while informative, can be improved for clarity and conciseness. Here are some specific suggestions:

**Clearer Structure:** Begin with a concise statement on the issue of terminological comparability and the need for standardized classifications in service research. Follow with a brief explanation of the study's focus on education services and its alignment with the PECUNIA project.**Streamlined Methodology Explanation:** Condense the explanation of the ontoterminology approach, focusing on the key steps without overwhelming detail. Emphasize the importance of accurate identification and disambiguation of services.**Results Highlight:** Clearly state the key findings, such as the low percentage of accurate services after disambiguation. This will help readers quickly grasp the study's outcomes.**Impact Statement:** Conclude with a concise statement on the significance of standardized methodologies and classifications in promoting comparability across sectors and ensuring a nuanced understanding of diverse services.**Revised Keywords:** Ensure that the keywords reflect the main concepts of the abstract, such as disambiguation, education services, service research, and the DESDE system.

**Introduction section**

The introduction provides a comprehensive overview of service science and its multidisciplinary nature, emphasizing the interconnectedness of various service sectors. Consider streamlining the introduction for better readability:

**Concise Background:** Condense the background information on service science and its multidisciplinary nature, focusing on its relevance to understanding complex interventions.**Clarity on Systems Thinking:** Clearly articulate the concept of systems thinking and its application to service provision, giving examples without delving too deeply into specific empirical studies.**Streamlined Structure:** Organize the introduction into shorter paragraphs, each addressing a specific aspect of service science, systems thinking, and the need for operational classifications. This will enhance overall clarity and engagement.**Transition to the DESDE System:** Introduce the DESDE system and its role in providing a standardized classification for services, setting the stage for the study's focus on education services.

**The methods section **

It provides a comprehensive overview of the study's design, context, and application of the DESDE system. However, it can benefit from some improvements for clarity and conciseness:

** Alignment with PECUNIA Project Objectives:**

   Ensure a more explicit connection between the study's methodology and the broader objectives of the PECUNIA project. Clearly state how the study contributes to the overarching goals of the project, providing a seamless transition from project context to the study design.

**Axes in the DESDE System:**

   Offer a more comprehensive explanation of the axes in the DESDE system. While a multiaxial code structure is mentioned, briefly describe each axis to aid readers unfamiliar with DESDE, enhancing overall clarity and understanding.

**Expert Involvement Elaboration:**

   Elaborate on the specific contributions of each expert working group (A, B, C). Specify the unique perspectives or expertise each group brought to the study, providing a more nuanced understanding of expert involvement and its impact on the classification process.

**Qualitative Data Integration:**

   If applicable, consider integrating qualitative data or quotes from expert consultations to enrich the narrative. This could offer a more vivid portrayal of the challenges faced and solutions proposed during the expert consultation process, enhancing the depth of the methodological description.

**Contextual Clarification:**

   Provide a brief, clear explanation of the PECUNIA project and its objectives before delving into the study design. This will help readers understand the broader context of the research and appreciate the study's relevance to the project's overarching goals.

**Conciseness in Study Design:**

   Streamline the description of the study design, emphasizing its alignment with the PECUNIA project. Highlight specific mental health disorders chosen for illustration without unnecessary detail, ensuring a balance between clarity and conciseness.

**Clarity in DESDE Application:**

   Clearly state the rationale for applying the DESDE classification to education services and explain how the coding process was conducted using a generic list of prototype services. Ensure that the application of DESDE is presented with clarity and relevance to the study.

**Simplified Education Contexts:**

   Condense detailed descriptions of education contexts in Spain and The Netherlands to focus on key aspects relevant to the study. Make the information more accessible to readers without overwhelming them with unnecessary details.

**Streamlined Governance Structure:**

   Streamline governance structure information for both countries, emphasizing key points relevant to education services. Ensure that the governance structure is presented in a clear and concise manner.

**Professional Qualifications Overview:**

    Briefly outline the qualifications and roles of key professionals in the education sector in Spain and The Netherlands. Ensure clarity and relevance to the study without delving into unnecessary details.

**DESDE System Explanation:**

    Provide a succinct yet comprehensive explanation of the DESDE system, its axes, and how it was utilized in the study. Include a concise description of the multiaxial code structure for a clearer understanding.

**Procedure Overview Streamlining:**

    Streamline the description of the assessment procedure, emphasizing key steps involved in service identification, disambiguation, and code development. Ensure clarity and simplicity in presenting the procedural overview.

**Expert Involvement Clarification:**

    Clearly outline the roles of the three working groups (A, B, C), providing a concise overview of their contributions to the study. Avoid redundancy and ensure that each group's role is distinctly communicated.

**Step-by-Step Process Breakdown:**

Break down the multistep process into clear, sequential steps, each with a succinct description. Ensure clarity in terminology, such as the distinction between disambiguation and development of prototype codes, for a more accessible presentation.

**Results section**

**Lack of Clarity in Disambiguation Criteria:**

The criteria for disambiguating services appear somewhat subjective and may benefit from a more explicit and standardized approach. Providing clearer guidelines for categorizing items as accurate, ambiguous, vague, or confusing could enhance the transparency and reproducibility of the classification process. Clearly articulate the criteria for classifying items into accurate, ambiguous, vague, or confusing categories. Consider providing illustrative examples to guide readers and potential future researchers in understanding the disambiguation process.

**Limited Exploration of Ambiguous and Vague Categories:**

The Results Section could delve deeper into the ambiguous and vague categories, providing more detailed examples and insights into why certain items fell into these classifications. A more in-depth analysis of these categories could shed light on the intricacies of classifying educational services. Dedicate more space to discussing items categorized as ambiguous or vague. Offer nuanced explanations for why certain items fell into these categories, providing contextual details that highlight the challenges faced during the classification process.

**Insufficient Exploration of Expert Suggestions:**

While expert opinions are briefly mentioned, the Results Section could benefit from a more comprehensive exploration of how expert feedback influenced the classification process. Explicitly linking expert comments to specific items in the disambiguation process would provide a clearer understanding of the impact of expert validation. Integrate expert comments more seamlessly into the narrative. Explicitly connect expert suggestions to specific items in the disambiguation process, emphasizing the role of expert validation in refining the classification criteria.

**Inconsistent Terminology:**

The use of terms like "accurate," "ambiguous," "vague," and "confusing" could be standardized throughout the section for better coherence. Ensuring consistent language will improve the overall readability and interpretation of the results. Ensure consistent use of terminology throughout the Results Section. Streamline the language used to describe the classification outcomes, making it easier for readers to follow and interpret the findings.

**The Discussion Section**

**Limited Exploration of Expert Suggestions:**

While the discussion briefly touches upon expert suggestions, it could benefit from a more detailed examination of how expert consultations influenced the refinement of service descriptions and the application of DESDE codes. Explicitly linking expert feedback to specific challenges or improvements in the classification system would enhance the discussion. Dedicate a section to thoroughly explore the impact of expert consultations on the classification system. Provide specific examples of services that were refined based on expert input and discuss how this collaboration improved the accuracy of the classification.

**Insufficient Exploration of Service Accuracy Levels:**

The discussion provides the overall percentage of services accurate at different levels of disambiguation, but a deeper analysis of the implications and challenges associated with these accuracy levels is warranted. Discussing specific examples or patterns in services that were accurate or faced challenges at different levels would provide a richer understanding. Conduct a more nuanced analysis of the accuracy levels reported, discussing specific services that excelled or faced challenges at different stages of disambiguation. This would provide a more comprehensive understanding of the classification process.

**Limited Exploration of Code Adaptation Suggestions:**

The discussion briefly mentions suggestions for adapting the DESDE system to the education sector, such as the need for new codes to differentiate day and evening school. Further exploration and discussion of these suggested adaptations, including potential implications and challenges, would enhance the completeness of the discussion. Elaborate on the suggestions for adapting the DESDE system to the education sector. Discuss the rationale behind proposed changes, potential benefits, and challenges associated with incorporating new codes or modifying existing ones.

**Incomplete Integration of ISCED System:**

The discussion mentions the potential complementarity of international classifications like DESDE and ISCED but does not delve into the practical implications of integrating these systems. Further exploration of how combining information from DESDE and ISCED could enhance the understanding of the education sector would strengthen the discussion. Explore in detail how integrating information from the DESDE and ISCED systems can provide a more holistic view of the education sector. Discuss specific examples where this combination could enhance the classification and understanding of educational services.

**Practical Implications: **

Expand on the practical implications of the study's findings for policymakers, educators, or researchers. Discuss how the standardized classification system could contribute to more effective educational service planning and delivery.

We look forward to receiving your revised manuscript.

Kind regards,

Faten Amer, PhD in Health Sciences , MBA, Pharmacy

Academic Editor

PLOS ONE

Journal Requirements:

"The PECUNIA project has received funding from the European Union’s Horizon 2020 research and innovation programme under grant agreement No 779292. 

This study also received financial support by a grant from the Carlos III Health Institute (PI18/01521). The funding agreement ensured the authors’ independence in designing the study, interpreting the data, and writing and publishing the report."

Additional Editor Comments: 

Editor’s comments:

After a thorough examination of your manuscript, it is apparent that your work exhibits significant potential. I kindly request some minor revisions to further enhance the manuscript. These suggestions are tailored to pinpoint specific areas where refinement could enhance overall clarity and amplify the impact of your work. The detailed suggestions are provided below, and I kindly ask you to respond to both the editor and the reviewers. Please address each suggestion individually in a separate file and highlight the modifications made in the manuscript. Your attention to these revisions is greatly appreciated.

Abstract

The abstract, while informative, can be improved for clarity and conciseness. Here are some specific suggestions:

1. Clearer Structure: Begin with a concise statement on the issue of terminological comparability and the need for standardized classifications in service research. Follow with a brief explanation of the study's focus on education services and its alignment with the PECUNIA project.

2. Streamlined Methodology Explanation: Condense the explanation of the ontoterminology approach, focusing on the key steps without overwhelming detail. Emphasize the importance of accurate identification and disambiguation of services.

3. Results Highlight: Clearly state the key findings, such as the low percentage of accurate services after disambiguation. This will help readers quickly grasp the study's outcomes.

4. Impact Statement: Conclude with a concise statement on the significance of standardized methodologies and classifications in promoting comparability across sectors and ensuring a nuanced understanding of diverse services.

5. Revised Keywords: Ensure that the keywords reflect the main concepts of the abstract, such as disambiguation, education services, service research, and the DESDE system.

Introduction section

The introduction provides a comprehensive overview of service science and its multidisciplinary nature, emphasizing the interconnectedness of various service sectors. Consider streamlining the introduction for better readability:

1. Concise Background: Condense the background information on service science and its multidisciplinary nature, focusing on its relevance to understanding complex interventions.

2. Clarity on Systems Thinking: Clearly articulate the concept of systems thinking and its application to service provision, giving examples without delving too deeply into specific empirical studies.

3. Streamlined Structure: Organize the introduction into shorter paragraphs, each addressing a specific aspect of service science, systems thinking, and the need for operational classifications. This will enhance overall clarity and engagement.

4. Transition to the DESDE System: Introduce the DESDE system and its role in providing a standardized classification for services, setting the stage for the study's focus on education services.

The methods section

It provides a comprehensive overview of the study's design, context, and application of the DESDE system. However, it can benefit from some improvements for clarity and conciseness:

1. Alignment with PECUNIA Project Objectives:

Ensure a more explicit connection between the study's methodology and the broader objectives of the PECUNIA project. Clearly state how the study contributes to the overarching goals of the project, providing a seamless transition from project context to the study design.

2. Axes in the DESDE System:

Offer a more comprehensive explanation of the axes in the DESDE system. While a multiaxial code structure is mentioned, briefly describe each axis to aid readers unfamiliar with DESDE, enhancing overall clarity and understanding.

3. Expert Involvement Elaboration:

Elaborate on the specific contributions of each expert working group (A, B, C). Specify the unique perspectives or expertise each group brought to the study, providing a more nuanced understanding of expert involvement and its impact on the classification process.

4. Qualitative Data Integration:

If applicable, consider integrating qualitative data or quotes from expert consultations to enrich the narrative. This could offer a more vivid portrayal of the challenges faced and solutions proposed during the expert consultation process, enhancing the depth of the methodological description.

5. Contextual Clarification:

Provide a brief, clear explanation of the PECUNIA project and its objectives before delving into the study design. This will help readers understand the broader context of the research and appreciate the study's relevance to the project's overarching goals.

6. Conciseness in Study Design:

Streamline the description of the study design, emphasizing its alignment with the PECUNIA project. Highlight specific mental health disorders chosen for illustration without unnecessary detail, ensuring a balance between clarity and conciseness.

7. Clarity in DESDE Application:

Clearly state the rationale for applying the DESDE classification to education services and explain how the coding process was conducted using a generic list of prototype services. Ensure that the application of DESDE is presented with clarity and relevance to the study.

8. Simplified Education Contexts:

Condense detailed descriptions of education contexts in Spain and The Netherlands to focus on key aspects relevant to the study. Make the information more accessible to readers without overwhelming them with unnecessary details.

9. Streamlined Governance Structure:

Streamline governance structure information for both countries, emphasizing key points relevant to education services. Ensure that the governance structure is presented in a clear and concise manner.

10. Professional Qualifications Overview:

Briefly outline the qualifications and roles of key professionals in the education sector in Spain and The Netherlands. Ensure clarity and relevance to the study without delving into unnecessary details.

11. DESDE System Explanation:

Provide a succinct yet comprehensive explanation of the DESDE system, its axes, and how it was utilized in the study. Include a concise description of the multiaxial code structure for a clearer understanding.

12. Procedure Overview Streamlining:

Streamline the description of the assessment procedure, emphasizing key steps involved in service identification, disambiguation, and code development. Ensure clarity and simplicity in presenting the procedural overview.

13. Expert Involvement Clarification:

Clearly outline the roles of the three working groups (A, B, C), providing a concise overview of their contributions to the study. Avoid redundancy and ensure that each group's role is distinctly communicated.

14. Step-by-Step Process Breakdown:

Break down the multistep process into clear, sequential steps, each with a succinct description. Ensure clarity in terminology, such as the distinction between disambiguation and development of prototype codes, for a more accessible presentation.

Results section

1. Lack of Clarity in Disambiguation Criteria:

The criteria for disambiguating services appear somewhat subjective and may benefit from a more explicit and standardized approach. Providing clearer guidelines for categorizing items as accurate, ambiguous, vague, or confusing could enhance the transparency and reproducibility of the classification process. Clearly articulate the criteria for classifying items into accurate, ambiguous, vague, or confusing categories. Consider providing illustrative examples to guide readers and potential future researchers in understanding the disambiguation process.

2. Limited Exploration of Ambiguous and Vague Categories:

The Results Section could delve deeper into the ambiguous and vague categories, providing more detailed examples and insights into why certain items fell into these classifications. A more in-depth analysis of these categories could shed light on the intricacies of classifying educational services. Dedicate more space to discussing items categorized as ambiguous or vague. Offer nuanced explanations for why certain items fell into these categories, providing contextual details that highlight the challenges faced during the classification process.

3. Insufficient Exploration of Expert Suggestions:

While expert opinions are briefly mentioned, the Results Section could benefit from a more comprehensive exploration of how expert feedback influenced the classification process. Explicitly linking expert comments to specific items in the disambiguation process would provide a clearer understanding of the impact of expert validation. Integrate expert comments more seamlessly into the narrative. Explicitly connect expert suggestions to specific items in the disambiguation process, emphasizing the role of expert validation in refining the classification criteria.

4. Inconsistent Terminology:

The use of terms like "accurate," "ambiguous," "vague," and "confusing" could be standardized throughout the section for better coherence. Ensuring consistent language will improve the overall readability and interpretation of the results. Ensure consistent use of terminology throughout the Results Section. Streamline the language used to describe the classification outcomes, making it easier for readers to follow and interpret the findings.

The Discussion Section

1. Limited Exploration of Expert Suggestions:

While the discussion briefly touches upon expert suggestions, it could benefit from a more detailed examination of how expert consultations influenced the refinement of service descriptions and the application of DESDE codes. Explicitly linking expert feedback to specific challenges or improvements in the classification system would enhance the discussion. Dedicate a section to thoroughly explore the impact of expert consultations on the classification system. Provide specific examples of services that were refined based on expert input and discuss how this collaboration improved the accuracy of the classification.

2. Insufficient Exploration of Service Accuracy Levels:

The discussion provides the overall percentage of services accurate at different levels of disambiguation, but a deeper analysis of the implications and challenges associated with these accuracy levels is warranted. Discussing specific examples or patterns in services that were accurate or faced challenges at different levels would provide a richer understanding. Conduct a more nuanced analysis of the accuracy levels reported, discussing specific services that excelled or faced challenges at different stages of disambiguation. This would provide a more comprehensive understanding of the classification process.

3. Limited Exploration of Code Adaptation Suggestions:

The discussion briefly mentions suggestions for adapting the DESDE system to the education sector, such as the need for new codes to differentiate day and evening school. Further exploration and discussion of these suggested adaptations, including potential implications and challenges, would enhance the completeness of the discussion. Elaborate on the suggestions for adapting the DESDE system to the education sector. Discuss the rationale behind proposed changes, potential benefits, and challenges associated with incorporating new codes or modifying existing ones.

4. Incomplete Integration of ISCED System:

The discussion mentions the potential complementarity of international classifications like DESDE and ISCED but does not delve into the practical implications of integrating these systems. Further exploration of how combining information from DESDE and ISCED could enhance the understanding of the education sector would strengthen the discussion. Explore in detail how integrating information from the DESDE and ISCED systems can provide a more holistic view of the education sector. Discuss specific examples where this combination could enhance the classification and understanding of educational services.

5. Practical Implications:

Expand on the practical implications of the study's findings for policymakers, educators, or researchers. Discuss how the standardized classification system could contribute to more effective educational service planning and delivery.

Reviewers' comments:

Reviewer's Responses to Questions

**Comments to the Author**

1. Is the manuscript technically sound, and do the data support the conclusions?

Reviewer #1: Yes

Reviewer #2: Partly

2. Has the statistical analysis been performed appropriately and rigorously? 

Reviewer #1: N/A

Reviewer #2: Yes

3. Have the authors made all data underlying the findings in their manuscript fully available?

Reviewer #1: Yes

Reviewer #2: Yes

4. Is the manuscript presented in an intelligible fashion and written in standard English?

Reviewer #1: Yes

Reviewer #2: Yes

5. Review Comments to the Author

Reviewer #1: This is a highly worth while study establishing a proof of concept in relation to the use of robust classification and terminology (taxonomical framework) to the area of school/education services for mental health service users with broader implications for systems analysis in education.

The authors have used existing an evidence based tools with expert analysis to assess the terminology used in education services and found considerable ambiguity and confusion. The outcomes of this work lay a basis for the development of standardised classification systems for comparison, planning and decision making in a key area of human services, namely education.

One minor point, the authors refer to three levels in the proc less of disambiguation - but there appeared to be four in the tables and discussion - namely accurate, vague, confusing and ambiguous. This needs to be clearer in the outline of "The Procedure" Step 2.

Reviewer #2: Dear Editor,

I hope this letter finds you well.

I would like to express my gratefulness for your trust and for giving me the opportunity to review the manuscript titled "Clarifying Terminology and Definitions in Education. no. PONE-D-23-36992 ".

According to objectives of the study "to test the applicability of a standard classification", and the Description and Evaluation of Services and DirectoriEs (DESDE), to evaluate education services for mental health users in the context of Spain and The Netherlands.

The title holds significant importance, as do the objectives of the study. However, here are some comments about where enhancements can be made to the manuscript.

1- Method need to clarify and demonstrate the structural formula employed in comparing the educational services between the two countries under examination. This clarification will provide a clearer understanding of the methodology employed and facilitate a more comprehensive analysis of our findings.

2- In the discussion part need to provide a detailed comparison between Spain and The Netherlands. This comparative analysis will light on any differences and contribute to a deeper understanding of the study’s outcomes.

Thank you once again for your consideration.

Sincerely,

Ahmad Hanani

6. PLOS authors have the option to publish the peer review history of their article (what does this mean?). If published, this will include your full peer review and any attached files.

Reviewer #1: No

Reviewer #2: No

---

## [Author Response · Author response to Decision Letter 0]

11 Jun 2024

Thank you for allowing us to review the paper for publication in the journal. We have carefully considered all the suggestions, and believe they were beneficial for crafting an enhanced version of the manuscript, which is now more succinct and clear, with examples and connections. 

To facilitate tracking changes in the response to reviewers, we have used blue colour coding (in the response to reviewers document) and indicated the corresponding pages and lines within the document -manuscript with track changes-.

Abstract

The abstract, while informative, can be improved for clarity and conciseness. Here are some specific suggestions:

1. Clearer Structure: Begin with a concise statement on the issue of terminological comparability and the need for standardized classifications in service research. Follow with a brief explanation of the study's focus on education services and its alignment with the PECUNIA project.

2. Streamlined Methodology Explanation: Condense the explanation of the ontoterminology approach, focusing on the key steps without overwhelming detail. Emphasize the importance of accurate identification and disambiguation of services.

3. Results Highlight: Clearly state the key findings, such as the low percentage of accurate services after disambiguation. This will help readers quickly grasp the study's outcomes.

4. Impact Statement: Conclude with a concise statement on the significance of standardized methodologies and classifications in promoting comparability across sectors and ensuring a nuanced understanding of diverse services.

5. Revised Keywords: Ensure that the keywords reflect the main concepts of the abstract, such as disambiguation, education services, service research, and the DESDE system.

Thank you for the suggestion, the abstract has been revised according to indications to clearly and concisely inform readers about the study

In the wake of the mental health crisis in children and adolescents, the coordination of education and mental health services has become a global priority. However, differing terminologies and classifications across sectors, hinder effective comparison. The classification in education focuses mainly on outputs like qualifications or throughputs like teaching programs. This proof-of-concept study tested the applicability of a standard classification of health services, the Description and Evaluation of Services and DirectoriEs (DESDE), to evaluate education services for mental health users in the context of Spain and The Netherlands. It was conducted alongside the PECUNIA project, that sought to develop methods for the assessment of mental health costs and outcomes in different sectors. The study followed an ontoterminology approach involving: 1) identification of services from a predefined list of 46 resource-use items, 2) Disambiguation of identified services with the DESDE, and classifying them as accurate, ambiguous, vague or confuse; and 3) external validation by an expert panel. The analysis was conducted at the level of type of resource, target population and care provision. From the initial list, only ten of the resources could be categorized as services using DESDE, and not activities, interventions or professionals. Only four of them (8,65%) were accurate across all disambiguation categories. Experts were unaware of terminology problems in classification of service provision in the education sector. Classifications and glossaries can clarify service naming, description and costing allowing comparative effectiveness analysis and facilitating cross-sectoral planning. This should be grounded in common methodologies, tools, and units of analysis. 

 Keywords: Disambiguation, education services, service research, DESDE system, PECUNIA project Introduction section

The introduction provides a comprehensive overview of service science and its multidisciplinary nature, emphasizing the interconnectedness of various service sectors. Consider streamlining the introduction for better readability:

1. Concise Background: Condense the background information on service science and its multidisciplinary nature, focusing on its relevance to understanding complex interventions.

2. Clarity on Systems Thinking: Clearly articulate the concept of systems thinking and its application to service provision, giving examples without delving too deeply into specific empirical studies.

3. Streamlined Structure: Organize the introduction into shorter paragraphs, each addressing a specific aspect of service science, systems thinking, and the need for operational classifications. This will enhance overall clarity and engagement.

4. Transition to the DESDE System: Introduce the DESDE system and its role in providing a standardized classification for services, setting the stage for the study's focus on education services.

 Thank you for providing a very useful guide to organize the introduction. The whole section has been amended considering suggestions, and structured in different paragraphs separating service science, systems thinking and operational classifications. The first paragraph has been slightly modified to strengthen the importance of an integrated perspective of service provision. A new reference has been incorporated. 

Pg3 lines 56-58. In the wake of the mental crisis in children and adolescents, the coordination of education and mental health services has become a global priority. Integrated youth services which rely on evidence-based models, are designed to provide effective care through a holistic approach (1). This integration is part of a broader discipline known as service science that has emerged in response to the growing complexity of the service sector, particularly in more developed economies (2).

1) Richards MC, Benson NM, Kozloff N, Franklin MS. Remodeling Broken Systems: Addressing the National Emergency in Child and Adolescent Mental Health. Psychiatr Serv [Internet]. 75(3):291–3. Available from: https://click.endnote.com/viewer?doi=10.1176%2Fappi.ps.20220283&token=WzIzMTQ4MTEsIjEwLjExNzYvYXBwaS5wcy4yMDIyMDI4MyJd.ksoQBjvAF3nehdR3zZhpRofSlgM

For the sake of clarity an example has been provided on how whole systems operate 

Pg3 lines 68-72 ‘For example, improvements in health services may enhance educational outcomes by increasing student mental health, wellbeing etc. and conversely educational advancements can lead to better health awareness’.

Explanation of the DESDE system has been improved and moved to the end of the section, to facilitate the understanding of the classification and the connection with the objectives of the study.

Pg 6 lines 134- 143. ‘The Description and Evaluation of Services and DirectoriEs (DESDE) is an example of an international, intersectoral classification system valid for the evaluation of complex systems (11). It originates from the field of mental health services research. In the past 25 years, it has been extended to incorporate other care sectors such as social (12), child and adolescent care, drug and alcohol, ageing, disabilities, long-term care (11,13). DESDE has been validated and extensively used in the field of health services research and health economics for the comparison of resource utilization across different settings, efficiency analysis and the calculation of unit costs, among others (14). Furthermore, DESDE is the only existing tool that provides local, bottom-up information that can be used across different sectors’.

The methods section 

It provides a comprehensive overview of the study's design, context, and application of the DESDE system. However, it can benefit from some improvements for clarity and conciseness:

1. Alignment with PECUNIA Project Objectives:

 Ensure a more explicit connection between the study's methodology and the broader objectives of the PECUNIA project. Clearly state how the study contributes to the overarching goals of the project, providing a seamless transition from project context to the study design.

The paragraph regarding the PECUNIA project has been reviewed to clarify the connection and synergy between both studies.

Pg7 lines 163-175 ‘The study was conducted alongside the ProgrammE in Costing, resource use measurement and outcome valuation for Use in multi-sectoral National and International health economic evaluAtions (PECUNIA) project (https://www.pecunia-project.eu/). PECUNIA was a European-funded consortium aimed at developing new standardised, harmonised and validated methods and tools for the assessment of costs and outcomes in European healthcare systems from a whole-system perspective including health and social care, employment, education and justice sectors. Although methods and tools developed within the PECUNIA project were meant to be generic, three mental health disorders (depression, post-traumatic stress disorder, and schizophrenia) were selected as illustrative examples. Building on the foundational goals of PECUNIA, this study’s focus on disambiguation and classification was related to the critical objectives of definition and harmonization within the project (11).The initial list of education resource-use items was compiled from literature reviews conducted within PECUNIA. Once classified with DESDE, these services were used in PECUNIA for cost calculation’.

2. Axes in the DESDE System:

 Offer a more comprehensive explanation of the axes in the DESDE system. While a multiaxial code structure is mentioned, briefly describe each axis to aid readers unfamiliar with DESDE, enhancing overall clarity and understanding.

A complete description of the axes has been included in the text

Pg 13 lines 286-296. ‘DESDE provides a multiaxial system that contains information on the following aspects of a service or a BSIC: a) the sector cluster (health, social, education, employment, justice); b) the target population, Including the age (e.g. CX – children and adolescents), gender, and the diagnosis group for which the service is intended. Code is based on the International Classification of Diseases 10th edition (ICD-10) (40)(e.g. F7-F8 - Mental retardation, disorders of psychological development), and the International Classification of Functioning, Disability and Health (ICF) (41); c) a code of the main service function or type of care and its additional qualifiers. This code is the core component of DESDE (Fig 1). Two DESDE axes were used for the disambiguation study, target population and service type of care.’

3. Expert Involvement Elaboration:

 Elaborate on the specific contributions of each expert working group (A, B, C). Specify the unique perspectives or expertise each group brought to the study, providing a more nuanced understanding of expert involvement and its impact on the classification process.

The contributions of each expert group has been explained and incorporated into their corresponding subsection within the procedure. 

Pg 14 lines 302-360: 

‘Procedure

This ontoterminology study followed a multistep process to assess the clarity of terms in education services and involved three working groups:

Step 1 - Service identification

Group A comprised members from Maastricht University (IP, LJ, SE, AP, RD). They undertook peer-reviewed and grey literature searches, as well as multi-national expert surveys that took place in six European countries prior to this study, as part of the PECUNIA project (25). (…)

Step 2 - Disambiguation 

Group B included two experts in disambiguation using DESDE (MR, LS) that analysed the list compiled by group A. (…) The process involved two levels:

- 1. Defining the unit of analysis (type of resource): group B reviewed the general list and ensured that the items and their descriptions corresponded to the same unit of analysis (commensurability), in this case ‘services’ and not to other units of analysis such as interventions, professionals, or outcomes. (…) 

2. DESDE classification: group B assessed information about the target population - including age and diagnosis group-, as well as the type of care provided in the services. (…)

Step 3- External validation

To finalize the disambiguation process and evaluate the suitability of DESDE for classifying education services, group C was consulted. This group consisted of a panel of external experts recruited from the current study’s co-authors’ professional network using purposive sampling (two from Spain, and three from the Netherlands). It included two academics, one public agent from the regional Ministry of Education and two third-sector professionals (AG, AR, TS, IO, MS). Their primarily role was to validate the services coded by group B, using their deep knowledge of the sector.

 Consultation involved individual semi-structured interviews. Experts were guided through a series of generic inquiries about education within the contexts of Spain and The Netherlands. Interview questions were pre-defined by the co-authors of the current study. (S1 Table 1). Experts critically reviewed the comprehensiveness and accuracy of the service list and verified the correctness of the assigned codes, taking into account the educational realities in both countries. 

4. Qualitative Data Integration:

 If applicable, consider integrating qualitative data or quotes from expert consultations to enrich the narrative. This could offer a more vivid portrayal of the challenges faced and solutions proposed during the expert consultation process, enhancing the depth of the methodological description.

This information has been included as part of the results of the study 

5. Contextual Clarification:

 Provide a brief, clear explanation of the PECUNIA project and its objectives before delving into the study design. This will help readers understand the broader context of the research and appreciate the study's relevance to the project's overarching goals.

This suggestion has been integrated in response to point 1. 

6. Conciseness in Study Design:

 Streamline the description of the study design, emphasizing its alignment with the PECUNIA project. Highlight specific mental health disorders chosen for illustration without unnecessary detail, ensuring a balance between clarity and conciseness.

The section addressing the study design has been deleted, the information previously included in this section has been move to the beginning of the method.

Pg8 lines157-162 ‘The study was designed as a proof-of-concept to test the applicability of the DESDE classification system to the education sector. It aimed to assess the clarity of service terms in this sector using an ontoterminology approach. Meaning the study of disambiguation of technical and scientific terms using standard classifications (26). This method helps identify precise definitions and relationships between terms, enhancing the retrieval of domain-specific information. It also aids in creating controlled vocabularies, classifications, and glossaries (27)’.

7. Clarity in DESDE Application:

 Clearly state the rationale for applying the DESDE classification to education services and explain how the coding process was conducted using a generic list of prototype services. Ensure that the application of DESDE is presented with clarity and relevance to the study.

Rationale for the application of the DESDE system has been included in the study design.

Pg 14, lines 299-303. ‘In this proof of concept study, the rationale for applying DESDE classification arose from the need to create a standardized framework for describing and assessing education services in a comparable manner. DESDE was originally developed for use in health and social care sectors. It’s methodological rigor made it ideal to bring similar benefits of standardization to the education sector’. 

8. Simplified Education Contexts:

 Condense detailed descriptions of education contexts in Spain and The Netherlands to focus on key aspects relevant to the study. Make the information more accessible to readers without overwhelming them with unnecessary details.

The structure of educational contexts in Spain and in The Netherlands have been refined and reduced. 

Pg9 lines 195-223 In Spain, education is mandatory from 6 to 15 years (28). Pre-school education while not compulsory, is available for children from birth to 6 ye

---

## [Editor Report · Decision Letter 1]

20 Jun 2024

Clarifying Terminology and Definitions in Education services for Mental Health Users: A Disambiguation Study

PONE-D-23-36992R1

Dear Dr. Gutierrez-Colosia,

We’re pleased to inform you that your manuscript has been judged scientifically suitable for publication and will be formally accepted for publication once it meets all outstanding technical requirements.

Kind regards,

Faten Amer, PhD in Health Sciences

Academic Editor

PLOS ONE

Additional Editor Comments (optional):

The authors have done all the required revisions and the manuscript is ready for publication.
---

## [Editor Report · Acceptance letter]

24 Jun 2024

PONE-D-23-36992R1 

PLOS ONE

Dear Dr. Gutierrez-Colosia, 

I'm pleased to inform you that your manuscript has been deemed suitable for publication in PLOS ONE. Congratulations! Your manuscript is now being handed over to our production team.

Kind regards, 

on behalf of

Dr. Faten Amer 

Academic Editor

PLOS ONE